# Artificial Intelligence Enhances Diagnostic Flow Cytometry Workflow in the Detection of Minimal Residual Disease of Chronic Lymphocytic Leukemia

**DOI:** 10.3390/cancers14102537

**Published:** 2022-05-21

**Authors:** Mohamed E. Salama, Gregory E. Otteson, Jon J. Camp, Jansen N. Seheult, Dragan Jevremovic, David R. Holmes, Horatiu Olteanu, Min Shi

**Affiliations:** 1Division of Hematopathology, Mayo Clinic, Rochester, MN 55905, USA; mohamedesalama292@gmail.com (M.E.S.); otteson.gregory@mayo.edu (G.E.O.); seheult.jansen@mayo.edu (J.N.S.); jevremovic.dragan@mayo.edu (D.J.); olteanu.horatiu@mayo.edu (H.O.); 2Biomedical Imaging, Mayo Clinic, Rochester, MN 55905, USA; camp.jon@mayo.edu (J.J.C.); holmes.david3@mayo.edu (D.R.H.III)

**Keywords:** artificial intelligence, flow cytometry, chronic lymphocytic leukemia, minimal residual disease

## Abstract

**Simple Summary:**

Flow cytometric immunophenotyping is critical in detecting minimal residual disease (MRD) in patients with chronic lymphocytic leukemia (CLL). However, flow cytometric analysis is complicated and time-consuming. Herein, we evaluated the performance of a deep neural network (DNN) in detecting CLL MRD and whether it could improve the diagnostic workflow in a clinical laboratory setting. Our findings demonstrated that a hybrid DNN approach had high accuracy in detecting CLL MRD; it standardized the gating strategy and dramatically reduced gating time, and it could be fully integrated into the existing clinical laboratory.

**Abstract:**

Flow cytometric (FC) immunophenotyping is critical but time-consuming in diagnosing minimal residual disease (MRD). We evaluated whether human-in-the-loop artificial intelligence (AI) could improve the efficiency of clinical laboratories in detecting MRD in chronic lymphocytic leukemia (CLL). We developed deep neural networks (DNN) that were trained on a 10-color CLL MRD panel from treated CLL patients, including DNN trained on the full cohort of 202 patients (F-DNN) and DNN trained on 138 patients with low-event cases (MRD < 1000 events) (L-DNN). A hybrid DNN approach was utilized, with F-DNN and L-DNN applied sequentially to cases. “Ground truth” classification of CLL MRD was confirmed by expert analysis. The hybrid DNN approach demonstrated an overall accuracy of 97.1% (95% CI: 84.7–99.9%) in an independent cohort of 34 unknown samples. When CLL cells were reported as a percentage of total white blood cells, there was excellent correlation between the DNN and expert analysis [r > 0.999; Passing–Bablok slope = 0.997 (95% CI: 0.988–0.999) and intercept = 0.001 (95% CI: 0.000–0.001)]. Gating time was dramatically reduced to 12 s/case by DNN from 15 min/case by the manual process. The proposed DNN demonstrated high accuracy in CLL MRD detection and significantly improved workflow efficiency. Additional clinical validation is needed before it can be fully integrated into the existing clinical laboratory practice.

## 1. Introduction

The level of minimal residual disease or measurable residual disease (MRD) in chronic lymphocytic leukemia (CLL) patients following therapy has been established as an independent prognostic factor for both progression-free survival and overall survival [1,2,3]. Therapeutic approaches for CLL have dramatically improved over the recent years, with the introduction of very effective targeted therapies [4,5,6,7]. The MRD in CLL patients is increasingly being used as a surrogate endpoint to assess the efficacy of CLL therapy [8,9].

Flow cytometry (FC) is an indispensable tool for the diagnosis, classification and monitoring of hematologic neoplasms because it enables rapid identification and semi quantification of an abnormal cell population [10]. Currently, FC serves as a fundamental strategic tool to monitor MRD in several hematologic malignancies [11,12,13,14,15,16]. FC identification of CLL cells has conventionally relied upon “difference from normal” and “leukemia-associated immunophenotype” approaches. The presence of CD5 and CD23, dim expression of CD20, dim surface immunoglobulin light chain (κ or λ) restriction, and bright expression of CD200, compared to normal mature B cells, are the main distinguishing features of CLL cells [17,18,19,20,21]. Demonstration of CD19/CD5 co-expression along with additional markers provides added value, particularly in atypical cases, as well as improved sensitivity (10^−4^–10^−5^) needed for MRD testing [13,22]. In the clinical laboratory, FC typically relies on manual gating of cell populations and interpretation of cell distribution on two-dimensional dot plots, which require expertise by clinical laboratory staff and pathologists. FC application for MRD testing suffers from a myriad of limiting factors for widespread adoption, including the difficulty of standardization and the complexity of the data analysis stage, where the gating step constitutes its major bottleneck. Computer-assisted analysis of clinical FC data is emerging as an objective, accurate, and easy-to-use approach in the flow cytometry laboratory. 

Computational algorithms are capable of extracting subtle features in relevant FC data to make highly accurate predictions and enable fast and objective visualization of cell populations. Dimensionality reduction and clustering are two key computational methods for clinical FC data analysis. Clustering algorithms group cells based on similarity of expression patterns and the applicability of machine learning in FC to recapitulate manual gating was previously established using the critical assessment of population identification (FlowCAP) method [23]. Dimensionality reduction allows for two-dimensional visualization of the multidimensional FC data and reduces data size. Uniform manifold approximation and projection for dimension reduction (UMAP) is an example of dimensionality reduction [24,25,26]. The two approaches reported objectivity and accuracy in clinical FC analysis, mainly for cell classification [27]. Recently, automated flow cytometric MRD assessment has been reported in B lymphoblastic leukemia (B-ALL) and acute myeloid leukemia (AML) [25,28]. Through a supervised machine learning approach using a combination of multiple Gaussian Mixture Models (GMM) as a parametric density model, the automated ALL MRD assessment showed a high correlation with expert-based quantification (F1-scores > 0.5 in more than 95% of samples) [28]. A semi-supervised approach based on UMAP algorithm was deployed for AML MRD detection with a median F1-score of 0.794 [25].

Here we sought to develop a deep neural network (DNN) approach for an automated analysis pipeline with the human in the loop for cell classification and quantification of CLL MRD in a clinical laboratory setting.

## 2. Materials and Methods

### 2.1. Study Cohort and Flow Cytometry Analysis

The study was approved by the Mayo Clinic Institutional Review Board. For model training, validation, and testing, samples collected from 202 consecutive CLL patients status post-therapy between February 2020 to May 2021 were evaluated for the presence of MRD in peripheral blood and bone marrow specimens (herein referred to as the development cohort). An independent clinical evaluation cohort comprising 34 “unknown” specimens were used to evaluate the final proposed workflow that incorporated the trained DNN models; 9 of these specimens with high numbers of abnormal CLL cells were additionally used to perform serial dilution studies by spiking into 20 normal (CLL-free) peripheral blood or bone marrow specimens at targeted clonal percentages of 0.02%, 0.002%, and/or 0.001% of white blood cell events.

FC study was performed using a single-tube 10-color CLL MRD panel consisting of CD5-BV480, CD19-PE-Cy7, CD20-APC-H7, CD22-APC, CD38-APC-R700, CD43-BV605, CD45-PerCP-Cy5.5, CD200-BV421, Kappa-FITC, and Lambda-PE. Except for the Kappa and Lambda antibodies (Dako, Carpinteria, CA, USA), all other fluorochrome-conjugated antibodies were from BD Biosciences, San Jose, CA, USA. Specimens were processed and stained using a routine lyse/wash/stain procedure as previously described [29]. A total of 1.2 × 10^6^ events per analysis tube were acquired on a FACSLyric flow cytometer (BD Biosciences, San Jose, CA, USA), and data were processed using Infinicyt software (Cytognos, Salamanca, Spain).

### 2.2. “Ground Truth” Classification of CLL MRD

“Ground truth” gating of event classes and the classification of CLL MRD-positive and MRD-negative status was performed by one trained expert using Infinicyt software (Cytognos, Spain). The development cohort comprised 143 CLL MRD-positive samples and 60 CLL MRD-negative samples, while the independent clinical evaluation cohort comprised 25 CLL MRD-positive samples and 9 CLL MRD-negative samples. The minimum number of events that can reliably be used to define an immunophenotypic abnormal population of CLL cells is 20 [13,22]. The laboratory-established analytic sensitivity of the assay is 0.002%, using a minimum of 20 events to define an abnormal population out of 1 million analyzed events. Gated cell populations from each specimen were provided to the DNN as individual FCS files. Raw, uncompensated data were used based on prior investigations demonstrating the robustness of the DNN approach to the presence of spectral overlap and equivalent performance with both uncompensated and compensated data [30].

### 2.3. Deep Neural Network (DNN)

A fully connected DNN with an input layer accepting light scatter properties (3 channels) and fluorescent channels (10 channels) from the CLL MRD tube (13 total channels) followed by three fully connected hidden layers (64, 128, and 64 nodes) and an output layer comprising 14 classes (abnormal cells [i.e., CLL cells], aggregates, B cells, basophils/dendritic cells, blasts, debris, erythroblasts, granulocytes, hematogones, monocytes, NK cells, plasma cells, T cells, and unknown), was trained with an overall cost target of <0.08 (Python 3.7 using Tensorflow and Keras) [28]. The choice of 3 hidden layers was based on earlier development work using a different use case [30] and observations of another group that used principal components analysis to study the optimal number of hidden layers in a DNN [31]. Rectified linear unit (ReLU) and softmax activation functions were used with a categorical cross-entropy loss function and Adam optimizer for the training and optimization of the classifiers. Softmax activation function and categorical cross-entropy are standard techniques used for multi-class problems, where the classes are mutually exclusive [32]. ReLU enables the more efficient training of deeper networks, compared to the commonly-used activation functions before 2011, for instance, the sigmoid or the hyperbolic tangent. ReLU is six times faster than other well-known activation functions and is one of the most commonly used activation functions in DNNs [33]. Studies have also shown that using the Adam optimizer improves the performance of wide and deep neural networks [34]. Based on our earlier development work, dropout regularization did not optimize DNN performance, so dropout layers were not included (data not shown). The DNN was trained, validated, and tested on the FCS files using a randomly selected sample of total events (n = 256,772,844) in an 80:10:10% ratio. Table 1 reflects the relative proportions for populations of interest. 

The DNN was initially trained with the training data set (80% of randomly selected total events), which will be referred to as the full cohort DNN (F-DNN) (n = 205,423,330 events from 202 cases). For optimization of performance in cases with low MRD cell counts (MRD < 1000 events), a second DNN was trained with a subset of samples (n = 145,748,173 events from 138 cases) that had low MRD cell count, referred to as the “low-count DNN” (L-DNN).

### 2.4. Statistical Analysis

DNN inferences for the test events in the development cohort and the independent clinical evaluation cohort (“unknown” samples) were compared to the expert ground truth classification, and the DNN performance for each event class was evaluated using the area under the receiver operating curve (AUC), sensitivity (recall), specificity, positive predictive value (precision), negative predictive value, and F1 score (general quality of discriminator) [Python 3.7 and Microsoft R Open]. To aid in the interpretation of tables and figures, the following classes were combined into a single category called “Other cell categories”: basophils/dendritic cells, blasts, erythroblasts, granulocytes, monocytes, NK cells, and unknown events. Exact binomial confidence intervals (95%) are shown for proportions.

When comparing the number of abnormal CLL cells (# CLL events) at the individual case level using test events from the development cohort, it was necessary to upsample the events since test events represented approximately only 10% of all case events. For upsampling, the number of abnormal CLL cells was first expressed as a percentage of total white blood cells (WBC) and then normalized to a total case WBC count of 1 million, which is the typical number of WBCs retained after removal of time errors, aggregates and debris.

Robust non-parametric (Passing-Bablok) regression was used to compare the DNN inferences for a total number of abnormal CLL events (# CLL events) and a percentage of abnormal CLL events of total white blood cells (% WBCs) with the expert analysis [Microsoft R Open]. Infinicyt software was used for visual comparison of populations of interest between the DNN inference and the expert gating. For the serially diluted samples, the proposed workflow was utilized to determine the absolute error (delta) for cases with CLL MRD events < 1000, and the proportionate error (% error) for cases with CLL MRD events ≥ 1000, for the total abnormal clonal cell population inference per unknown sample versus the expert analysis. The level of significance (α) was chosen as 0.05.

## 3. Results

### 3.1. Initial DNN Training and Performance

The DNN was trained using 80% of events in the development cohort (F-DNN) and reached the pre-specified cost target (0.079) in 2281 epochs. Population-specific performance metrics of the F-DNN for test events in the development cohort are shown in Table 1 and the confusion matrix of F-DNN inferences versus “ground truth” labels is shown in the Appendix A, Appendix A. Abnormal (CLL) cells were detected with a sensitivity of 99.3% and a specificity above 99.9%; the AUC for abnormal cells was greater than 0.999. The F-DNN demonstrated an AUC ≥ 0.996 for other cell classes of interest in the development cohort (Figure 1A).

At the sample/case level in the development cohort (test events only), F-DNN inferences showed excellent correlation with the expert abnormal CLL event counts (Pearson correlation coefficient, r > 0.99, Figure 2A), except at abnormal CLL event counts less than 1000 (r = 0.70, Figure 2B). Although the slope of the Passing–Bablok regression line was close to 1 (slope = 1.001, 95% CI: 0.998–1.010), the intercept of 74.000 (95% CI: 60.831–93.835) was significant for cases with low abnormal CLL event counts.

### 3.2. DNN Optimization

For further optimization, a separate DNN with similar architecture to the F-DNN was trained using cases with CLL MRD events < 1000 (L-DNN). The training dataset included most of the cases (138 of 202) and a total of 145,748,173 events. The L-DNN trained network showed a higher specificity for abnormal cells (>99.9%) with a sensitivity of 80.8% for the test events in the development cohort. Population-specific performance metrics of the L-DNN for this subset of test events in the development cohort are shown in Table 2, and the confusion matrix of L-DNN inferences versus “ground truth” labels is shown in the Appendix A. Similar to the F-DNN, the L-DNN demonstrated an AUC ≥ 0.996 for all cell classes of interest in this subset of test events in the development cohort (Figure 1B). 

The L-DNN demonstrated an improved correlation with the expert analysis at abnormal CLL event counts less than 1000 (r = 0.96) (Figure 2C). For CLL MRD status designation, a hybrid DNN approach was utilized, whereby both the F-DNN and L-DNN were applied sequentially to cases; F-DNN was applied to all cases and L-DNN was applied only to cases with low-count MRD (<1000 events) predicted by F-DNN (Figure 3). This hybrid DNN workflow resulted in excellent correlation throughout the analyzed range (r > 0.99) with a Passing–Bablok regression slope of 0.998 (95% CI: 0.995–1.000) and the intercept of 0.000 (95% CI: 0.000–0.046) (Figure 2D).

### 3.3. Performance of Hybrid DNN Approach in Independent Clinical Evaluation Cohort

An independent cohort of 34 unknown samples was utilized for clinical evaluation of the final proposed workflow that incorporated the trained DNN models. These samples were randomly selected from 34 CLL patients that were collected between 2020 to 2022. The samples for flow cytometric testing were prepared by different technologists, run on three FACSLyric flow cytometers, and analyzed by three experts. A variety of lots of flow cytometric reagents, including antibodies, were used for testing over that time frame. Using an event threshold of 20 abnormal CLL events, the expert gating identified CLL MRD ranged from 0–98% with a median of 1.851%. The hybrid DNN approach demonstrated an overall accuracy of 97.1% (95% CI: 84.7–99.9%), with a sensitivity of 100.0% (95% CI: 86.3–100.0%), specificity of 88.9% (95% CI: 51.8–99.7%), positive predictive value of 96.2% (80.4–99.9%), and negative predictive value of 100% (63.1–100%) (Table 3).

To better demonstrate the hybrid DNN performance around the cutoff of 0.002%, serial dilution studies were performed by spiking CLL cells from nine patients into normal specimens at targeted CLL percentages of 0.02%, 0.002%, and 0.001% of analyzed white blood cells. Triplicates were performed on two different dilutions to establish reproducibility (Table 4). Overall, quantitative estimates of CLL MRD were similar between the hybrid DNN predictions and the “ground truth” case counts. Of note, the hybrid DNN approach, and the L-DNN in particular, failed to detect abnormal events in serial dilution samples of one of the nine CLL MRD-positive cases when the MRD level was low (case CLL-2). L-DNN classified all of the serial dilution samples for this case as CLL MRD-negative; further review identified that the abnormal events in this case (CLL-2 in Table 4) displayed an aberrant phenotype with brighter expression of CD20, which is an unusual immunophenotype for CLL. Although these events were not classified as abnormal CLL cells by the L-DNN, they were detected by the F-DNN.

When all clinical evaluation cases were included (34 unknown and 26 serial dilution samples), and abnormal CLL cells were reported as a percentage of total WBCs, there was an excellent correlation between the DNN inference and expert analysis (r > 0.999). The slope of the Passing–Bablok regression line was close to 1 (slope = 0.997, 95% CI: 0.988–0.999), and the intercept was close to 0 (intercept = 0.001, 95% CI: 0.000–0.001), demonstrating near-equivalence between both methods (Figure 4).

### 3.4. Process Efficiency and Analysis Time Considerations

Based on local data, the CLL MRD flow cytometric analysis time was estimated to be 15–20 min per case for an experienced technologist using commercially available gating/analysis tools. The hybrid DNN approach using a workstation with an Intel Core i5-6500 CPU @ 3.20 GHz and 8 GB of RAM reduced analysis time to approximately 12 s/case. The developed DNN workflow also permitted subsequent manual review of AI-gated events for further refinement of cell populations. Side-by-side comparison dot plots at the marker and event level for three representative samples with different CLL MRD levels are shown in Figure 5, which demonstrated visual similarity between manual expert gating and automated DNN gating. 

## 4. Discussion

Previous studies applied dimension reduction such as principal component analysis or clustering methods combined with machine learning in FC to demonstrate their utility in the diagnosis or classification of hematologic malignancies [29,30]. Although the successful applicability of an automated computational pipeline to identify CLL cells by multidimensional FC was previously illustrated [24,31], to the best of our knowledge, no published algorithm has investigated automated quantitative population analysis for CLL MRD.

Our findings support the feasibility and advantages of using DNN in a flow cytometry laboratory to detect CLL MRD. Our DNN model demonstrated its high accuracy in detecting CLL MRD with comparable performance to a gold standard (expert technologist). Using a minimum 20-abnormal-event threshold for CLL MRD positivity, the algorithmic F-DNN and L-DNN approach demonstrated 97.1% accuracy, 96.2% PPV, and 100% NPV in a clinical evaluation cohort. In addition, the F-DNN and L-DNN algorithm showed a strong correlation with an expert designation of CLL MRD in almost all serial dilution samples from 9 patients except for one case. In this case, the very low level of CLL MRD was detected by the F-DNN while being missed by the L-DNN. This is because the L-DNN was not exposed to the atypical CLL immunophenotype (brighter CD20 expression) due to limited CLL training events for L-DNN, in contrast to the F-DNN which was trained on a significantly larger number of CLL events with a greater spectrum of aberrant antigen expression. Nonetheless, the hybrid workflow demonstrated accuracy approaching 100% for CLL MRD cases with typical immunophenotypes. 

The explainable AI element in our proposed approach cannot be emphasized enough since it enables the user to visually evaluate, interpret, and trust the results. The proposed AI approach is particularly beneficial to the flow cytometry laboratory that utilizes clustering algorithms in the analysis software such as Infinicyt. This software automatically clusters events based on all parameters. However, assigning each cluster to the proper hematopoietic category is a time-consuming manual process, requires a substantial understanding of the immunophenotype of each cell type, and varies among users. Our model showed population-specific DNN gating had an excellent correlation to the expert gating, with almost all 14 cell classes achieving high sensitivity and specificity with AUC ≥ 0.99 for both F-DNN and the L-DNN. This allows the user to quickly review each population without manual gating. Thus, DNN gating standardizes the gating strategy and limits the variability of manual gating. DNN gating also dramatically reduces gating time to only 12 s/case, compared to 15 min/case by manual process. This approach would result in savings of 246.7 h of gating time for 1000 CLL MRD cases tested in our laboratory. In high-throughput reference laboratories like ours, this approach can be translated into significantly shorter turnaround time for results delivery and health care dollar savings; the relevance of both to value-based medicine cannot be understated. Even more broadly, this approach enables the expert-optimized algorithm to be implemented outside of academic clinical practice. We believe that a similar DNN approach can be applied to other hematologic malignancies where clonal cells demonstrate a distinct, reproducible aberrant immunophenotype, such as in multiple myeloma MRD analysis.

The design of the DNN allows secondary review of the assigned population, giving users flexibility with AI classification. This study demonstrated that flow cytometry laboratories could place the proposed algorithmic DNN analysis into existing clinical practice with minor changes to the current workflow. After F-DNN analysis, if CLL events ≥ 1000, the user could rapidly review the immunophenotype of the CLL events from dot plots and report the result as positive once confirmed. If CLL events < 1000, the case will be automatically analyzed by the L-DNN. If 20 ≤ CLL events < 1000, the user should review the dot plots of CLL events and, if necessary, other confounding classes such as polytypic B cells, hematogones, plasma cells, and T cells to ensure an accurate result. If CLL events < 20, in addition to reviewing the dot plots of CLL events and other confounding classes, the user could regate the case as appropriate. 

The current model has several limitations. The DNN model is panel-specific; any antibody modification (change antibody or fluorescence) could affect its performance. The DNNs are also immunophenotype specific, and do not currently perform well with non-CLL immunophenotype, since the DNNs have not been specifically trained to identify these immunophenotypes; this limitation could be readily overcome through further DNN training on border cases with non-CLL immunophenotype. In addition, the kappa and lambda immunoglobulin light chain ratio has not played an important role in the determination of CLL MRD based on the design of the DNN. Although this model proved the robustness of the approach to many pre-analytic variables, such as sample preparation and reagent usages, additional studies are also required to demonstrate whether there is performance drift over time. Finally, the DNN may potentially be further optimized by appropriately weighting an important cell class, such as abnormal CLL cell class, higher than other cell classes, by employing cost-sensitive learning or by synthetically upsampling abnormal CLL cells in cases < 1000 abnormal CLL cells.

## 5. Conclusions

The DNN-based gating workflows performed adequately for the clinical laboratory but outperformed the manual traditional clinical workflow providing shorter turnaround time and increased efficiency. Our approach of automated cell population gating and identification of abnormal cells, paired with secondary review of the assigned populations, paves a path for laboratories to place AI analysis into existing clinical practice as an adjunctive tool with minor changes to the current workflow. The automated DNN may be considered a highly standardized analysis technologist and generates FCS files that can be fully integrated into existing clinical laboratory practice. Our findings support the applicability of AI in CLL MRD FC clinical workflow and provide a framework for MRD FC testing for AML, ALL and MM, as well as other FC testing in hematologic disorders in the clinical laboratory. To our knowledge, this is the first study to prove the accuracy and efficiency of AI in the quantitative detection of CLL MRD, which broadens the application of AI in disease monitoring and prognostic prediction.

## Figures and Tables

**Figure 1 cancers-14-02537-f001:**
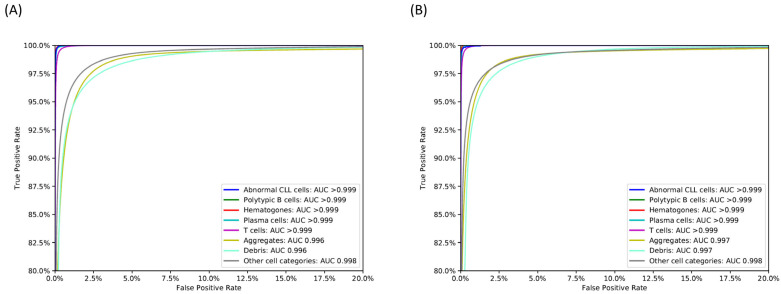
Receiver operating characteristic (ROC) curves for relevant cell classes in the test cohort for the (**A**) F-DNN network and (**B**) L-DNN network, with corresponding area under ROC (AUC). CLL: chronic lymphocytic leukemia. Other cell categories include granulocytes, monocytes, natural killer (NK) cells, blasts, erythroblasts, basophils, and dendritic cells; these populations were identified as individual classes by the DNN but combined into a single category for visualization purposes.

**Figure 2 cancers-14-02537-f002:**
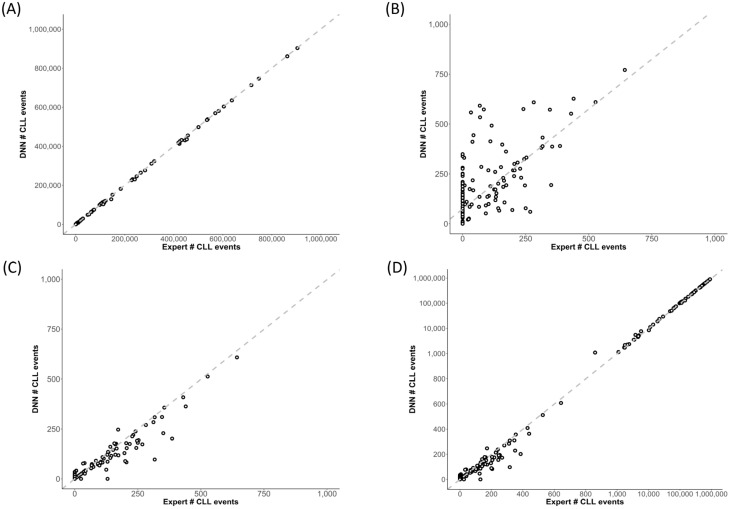
Scatterplots showing expert analysis and DNN inference of abnormal CLL events in the test cohort. (**A**,**B**) F-DNN inferences showed excellent correlation with the expert abnormal CLL event counts (Pearson correlation coefficient, r > 0.99); (**A**), except at abnormal CLL event counts less than 1000 (r = 0.70). (**B**) The dashed line represents the Passing–Bablok regression line for the F-DNN inferences versus the expert “ground truth”: slope = 1.001 (95% CI: 0.998–1.010) and intercept = 74.000 (95% CI: 60.831–93.835). (**C**) The L-DNN showed an improved correlation with the expert “ground truth” at abnormal CLL event counts less than 1000 (r = 0.96). (**D**) Using the F-DNN inference if greater than or equal to 1000 abnormal events and the L-DNN inference if less than 1000 abnormal events resulted in excellent correlation throughout the analyzed range (r > 0.99) with a Passing–Bablok regression slope of 0.998 (95% CI: 0.995–1.000) and the intercept of 0.000 (95% CI: 0.000–0.046), shown in subplots (**C**,**D**).

**Figure 3 cancers-14-02537-f003:**
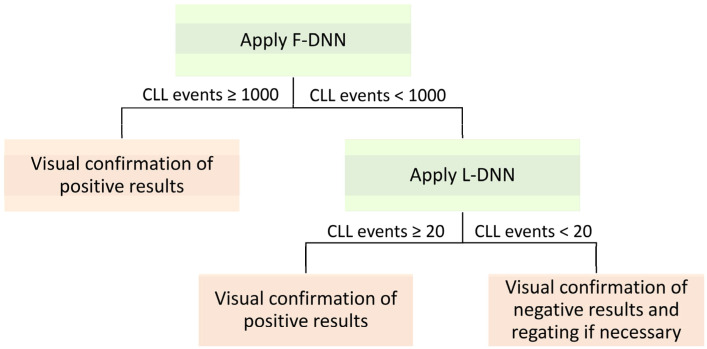
Proposed approach to apply F-DNN on all cases followed by L-DNN on cases with < 1000 abnormal events.

**Figure 4 cancers-14-02537-f004:**
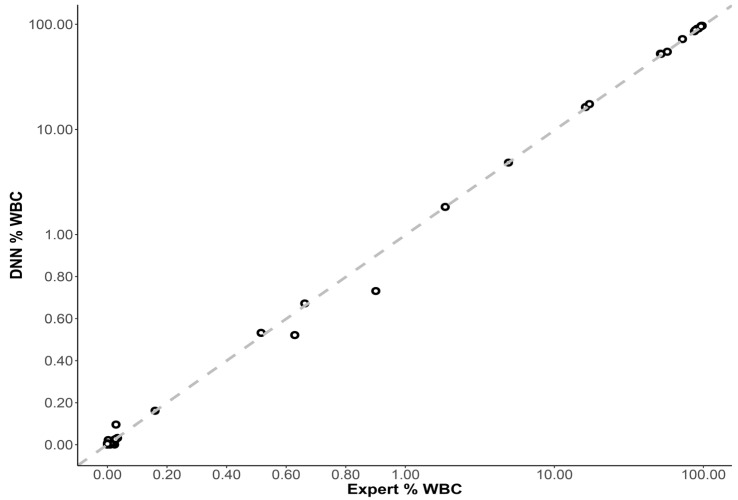
Scatterplot showing expert and DNN inference of abnormal CLL events as a % of total white blood cells (WBCs) for each unknown case (n = 60, including the serial dilution samples). The dashed line represents the Passing–Bablok regression line: slope = 0.997 (95% CI: 0.988–0.999) and intercept = 0.001 (95% CI: 0.000–0.001).

**Figure 5 cancers-14-02537-f005:**
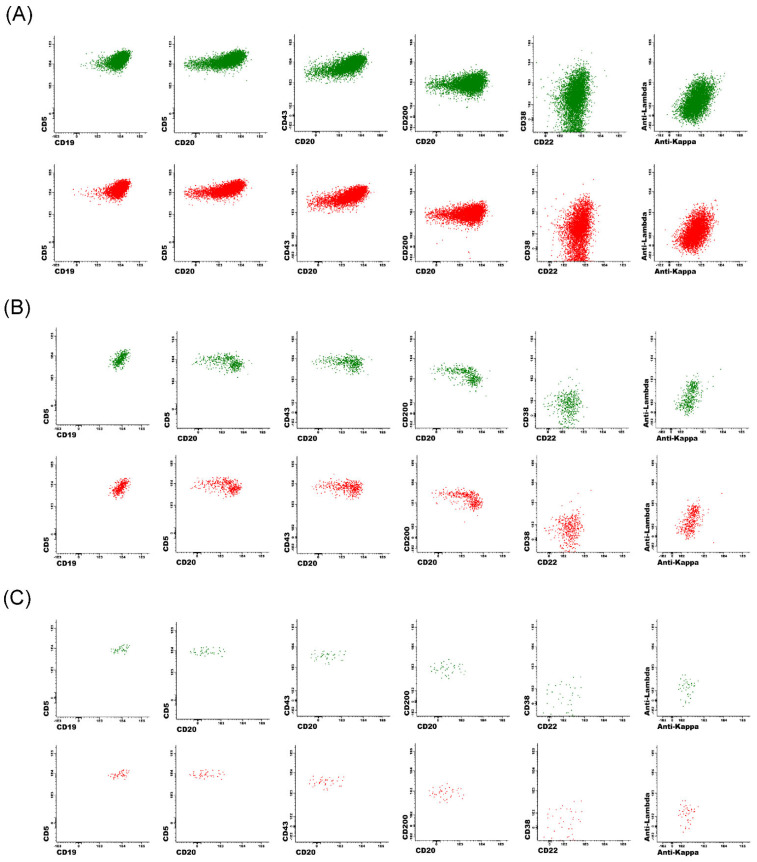
Side-by-side comparison of 2D dot plots show expert-gated CLL populations (green) and DNN-gated CLL populations (red) in three representative cases, using common marker combination. (**A**) 6076 expert-gated CLL events with a MRD level at 0.76% and 5147 DNN-gated CLL events with a MRD level at 0.73%. (**B**) 449 expert-gated CLL events with a MRD level at 0.026% and 424 DNN-gated CLL events with a MRD level at 0.031%. (**C**) 41 expert-gated CLL events with a MRD level at 0.0030% and 42 DNN-gated CLL events with a MRD level at 0.0035%.

**Table 1 cancers-14-02537-t001:** Performance metrics of the F-DNN for cell populations of interest using the test cohort (n = 25,671,598 events from 202 unique cases). Accuracy, sensitivity, specificity, PPV, and NPV are presented as dimensionless ratios.

Population	TP	TN	FP	FN	Accuracy	Sensitivity	Specificity	PPV	NPV	F1
CLL cells	1,316,803	24,331,391	13,539	9865	0.9991	0.9926	0.9994	0.9898	0.9996	0.9912
Polytypic B cells	195,925	25,464,627	6582	4464	0.9996	0.9777	0.9997	0.9675	0.9998	0.9726
Hematogones	93,642	25,569,274	5298	3384	0.9997	0.9651	0.9998	0.9465	0.9999	0.9557
Plasma cells	11,033	25,654,446	2914	3205	0.9998	0.7749	0.9999	0.7911	0.9999	0.7829
T cells	2,697,327	22,909,067	37,470	27,734	0.9975	0.9898	0.9984	0.9863	0.9988	0.9881
Aggregates	1,739,028	23,610,304	149,637	172,629	0.9874	0.9097	0.9937	0.9208	0.9927	0.9152
Debris	3,611,475	21,611,955	185,744	262,424	0.9825	0.9323	0.9915	0.9511	0.988	0.9416
Other cell Categories	15,297,918	9,841,675	307,263	224,742	0.9793	0.9855	0.9697	0.9803	0.9777	0.9829

**Table 2 cancers-14-02537-t002:** Performance metrics of the L-DNN for cell populations of interest using the test cohort (n = 18,212,266 events from 138 unique cases). Accuracy, sensitivity, specificity, PPV, and NPV are presented as dimensionless ratios.

Population	TP	TN	FP	FN	Accuracy	Sensitivity	Specificity	PPV	NPV	F1
CLL cells	1358	18,210,402	184	322	1.0000	0.8083	1.0000	0.8807	1.0000	0.8430
Polytypic B cells	177,419	18,027,561	4405	2881	0.9996	0.9840	0.9998	0.9758	0.9998	0.9799
Hematogones	70,908	18,135,313	3494	2551	0.9997	0.9653	0.9998	0.9530	0.9999	0.9591
Plasma cells	10,749	18,196,392	2426	2699	0.9997	0.7993	0.9999	0.8159	0.9999	0.8075
T cells	1,943,032	16,226,465	23,739	19,030	0.9977	0.9903	0.9985	0.9879	0.9988	0.9891
Aggregates	1,213,909	16,810,440	82,813	105,104	0.9897	0.9203	0.9951	0.9361	0.9938	0.9281
Debris	2,585,138	15,324,944	145,941	156,243	0.9834	0.9430	0.9906	0.9466	0.9899	0.9448
Other cell categories	11,750,260	6,094,852	196,491	170,663	0.9798	0.9857	0.9688	0.9836	0.9728	0.9846

**Table 3 cancers-14-02537-t003:** CLL MRD status designation comparison between expert-gated (ground truth) and DNN-gated cases in the independent validation cohort (34 unknown cases, excluding serial dilution samples).

		Expert	
DNN		CLL MRD−	CLL MRD+
	CLL MRD−	8	0
	CLL MRD+	1	25
Accuracy (%)		97.1 (84.7–99.9)	
Sensitivity (%)		100.0 (86.3–100.0)	
Specificity (%)		88.9 (51.8–99.7)	
PPV (%)		96.2 (80.4–99.9)	
NPV (%)		100.0 (63.1–100.0)	

PPV: positive predictive value; NPV: negative predictive value.

**Table 4 cancers-14-02537-t004:** CLL MRD comparison between expert-gated (ground truth) and DNN-gated from serially diluted samples (n = 9 unique cases). The absolute error (delta), for cases with CLL MRD events < 1000 and the proportionate error (% error), for cases with CLL MRD events ≥ 1000, for the total abnormal clonal cell population inference per unknown sample versus the “ground truth” population was determined using the proposed workflow described in Figure 3. For the CLL MRD % WBCs, the absolute difference is shown.

Case	Dilution	Expert CLL MRD # Events	DNN CLL MRD # Events	Absolute or Proportionate Error for # Events	Expert CLL MRD % WBCs	DNN CLL MRD % WBCs	Absolute Error for % WBCs
CLL-1	UD	1,132,442	1,096,836	−3.144%	88.878	88.521	−0.357
	0.02%	436	431	−5	0.022	0.023	0.001
	0.002%	61	60	−1	0.003	0.003	0.000
	0.001%	22	24	2	0.001	0.002	0.000
CLL-2	UD	1,303,375	1,116,751	−14.319%	92.812	91.454	−1.357
	0.02%	492	11	−481	0.024	0.001	−0.023
	0.002%	29	7	−22	0.002	0.000	−0.001
	0.001%	17	6	−11	0.009	0.000	−0.009
CLL-3	UD	725,221	715,315	−1.366%	97.025	96.949	−0.075
	0.02%	465	436	−29	0.034	0.032	−0.002
	0.002%	62	64	2	0.003	0.003	0.000
	0.001%	22	22	0	0.001	0.001	0.000
CLL-4	UD	627,879	622,034	−0.931%	51.493	52.536	1.042
	0.02%	327	322	−5	0.027	0.027	0.000
	0.002%	33	37	4	0.003	0.003	0.000
	0.002%	38	49	11	0.003	0.004	0.001
	0.002%	29	31	2	0.003	0.003	0.000
	0.001%	11	18	7	0.001	0.002	0.001
	0.001%	12	22	10	0.001	0.002	0.001
	0.001%	20	24	4	0.002	0.002	0.000
CLL-5	UD	558,396	555,029	−0.603%	51.997	52.340	0.343
	0.02%	368	1207	227.989%	0.029	0.096	0.067
	0.002%	28	73	45	0.002	0.005	0.003
CLL-6	UD	145,914	142,952	−2.03%	57.279	54.866	−2.413
	0.02%	324	128	−196	0.022	0.009	−0.013
	0.002%	45	16	−29	0.003	0.001	−0.002
CLL-7	UD	722,712	703,261	−2.691%	90.674	90.722	0.048
	0.002%	70	88	18	0.007	0.009	0.002
	0.001%	49	76	27	0.004	0.007	0.003
CLL-8	UD	1,139,331	1,067,634	−6.293%	96.586	96.289	−0.297
	0.02%	434	408	−26	0.025	0.023	−0.002
	0.002%	37	33	−4	0.002	0.002	0.000
CLL-9	UD	1,114,004	1,086,919	−2.431%	96.276	96.040	−0.236
	0.02%	451	358	−93	0.026	0.020	−0.005
	0.002%	48	41	−7	0.003	0.022	0.019

## Data Availability

Data is contained within the article and Appendix A.

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
