# Peer review of "Artificial Intelligence Enhances Diagnostic Flow Cytometry Workflow in the Detection of Minimal Residual Disease of Chronic Lymphocytic Leukemia"

_cancers, 2022, doi:10.3390/cancers14102537_

Round 1
Reviewer 1 Report
Assessing status and frequency of minimal residual disease (MRD) in leukemia patients is critical to patient outcomes. In this study, the authors investigate the ability of artificial intelligence (AI) to potentially improve efficiency in detecting MRD using chronic lymphocytic leukemia (CLL) samples. They developed deep neural networks (DNN) trained using a 10-color CLL MRD panel, including on the hard to analyze low event cases. They are able to show a high degree of replicable accuracy. In addition, analysis time was reduced 75-fold and can be done so without significantly changing existing clinical laboratory practice. Overall, this study will be of significant interest to the field. First, a few issues should be addressed:
In the Introduction, sufficient background is provided for MRD and its importance. However, for augmented intelligence, far too little background is provided.
For the results, a more thorough demonstration with side-by-side comparisons of a representative sample of "quick" augmented analysis vs. unassisted analysis would be helpful. The 2 may look very similar. If so, this comparison would help support the authors' conclusions.
The author's use the terms "augmented" and "artificial" intelligence. These terms are not completely interchangeable, however. The authors should be appropriately consistent as these are important definitions impacting the significance of their work.
Is this method applicable only to CLL or could this be applied more broadly. The authors could address this issue in the Discussion.
Author Response
We greatly appreciate the comments of the reviewers and believe that the suggested additions to our manuscript have served to further enhance its quality.
Review #1:
Assessing status and frequency of minimal residual disease (MRD) in leukemia patients is critical to patient outcomes. In this study, the authors investigate the ability of artificial intelligence (AI) to potentially improve efficiency in detecting MRD using chronic lymphocytic leukemia (CLL) samples. They developed deep neural networks (DNN) trained using a 10-color CLL MRD panel, including on the hard to analyze low event cases. They are able to show a high degree of replicable accuracy. In addition, analysis time was reduced 75-fold and can be done so without significantly changing existing clinical laboratory practice. Overall, this study will be of significant interest to the field. First, a few issues should be addressed:
In the Introduction, sufficient background is provided for MRD and its importance. However, for augmented intelligence, far too little background is provided.
Response: We have included additional relevant background on computational and automated flow cytometry on page 2, line 66-83.
For the results, a more thorough demonstration with side-by-side comparisons of a representative sample of "quick" augmented analysis vs. unassisted analysis would be helpful. The 2 may look very similar. If so, this comparison would help support the authors' conclusions.
Response: We have added additional examples of dot plots (figure 5) demonstrating visual similarity between manual expert gating and automated DNN gating.
The author's use the terms "augmented" and "artificial" intelligence. These terms are not completely interchangeable, however. The authors should be appropriately consistent as these are important definitions impacting the significance of their work.
Response: We have removed all mentions of “augmented intelligence” and instead have standardized the terminology by using “artificial intelligence”.
Is this method applicable only to CLL or could this be applied more broadly. The authors could address this issue in the Discussion.
Response: We believe that a similar DNN approach can be applied to other hematologic malignancies where clonal cells demonstrate a distinct, reproducible aberrant immunophenotype, such as in multiple myeloma MRD analysis. We have discussed this briefly on page 12, line 360-363.
Reviewer 2 Report
Authors contributions:
According to the authors, the limitations of flow cytometric (FC) immunophenotyping is that the method is critical but time-consuming in diagnosing minimal residual disease (MRD).
To solve some of the problems with FC, the authors have proposed deep neural networks (DNN) that were trained on a 10-color CLL MRD panel from treated CLL patients, including DNN trained on the full cohort of 202 patients (F-DNN), and DNN trained on 138 patients with low-event cases (MRD<1000 events) (L-DNN).
According to the results obtained, the hybrid DNN approach demonstrate an overall accuracy of 97% (95% CI: 85% - 99%) in an independent cohort of 34 samples, not used in the development of the DNN.
The reported correlation is strong (r> 0.99), between CLL cells as a percentage of total white blood cells, and the DNN and expert analysis.
The proposed method reduces the data processing time from 15 min to 12 s.
I have some reviewer notes:
Abstract. You have to show one sentence about how your work will be continued.
Introduction. This part is too short. You have to give more details about what authors have as a results in their works.
“2.3. Deep neural network (DNN)”. It is not clear how you select the sample size.
“2.4. Statistical analysis”. What is your level of significance?
Table 1. Are the accuracy levels dimensionless? You have to describe it in the text above the table. Also for Table 2.
Figure 2B. It is not clear why there is low correlation between dependent and independent variables in your linear model. Where is the problem?
Figure 4. Are all data in the confidence interval?
Discussion. You have to compare your results with minimum 3 other papers (More is better). Also, you have to show what is the accuracy of the results in other papers.
Conclusion. You have to show how your results improve the known solutions in this study area.
I have some suggestions:
It will be good to show the accuracy of the compared methods, presented from other authors. Also, make more comparative analyses. It will improve your contribution.
Author Response
We greatly appreciate the comments of the reviewers and believe that the suggested additions to our manuscript have served to further enhance its quality.
Review #2:
According to the authors, the limitations of flow cytometric (FC) immunophenotyping is that the method is critical but time-consuming in diagnosing minimal residual disease (MRD).
To solve some of the problems with FC, the authors have proposed deep neural networks (DNN) that were trained on a 10-color CLL MRD panel from treated CLL patients, including DNN trained on the full cohort of 202 patients (F-DNN), and DNN trained on 138 patients with low-event cases (MRD<1000 events) (L-DNN).
According to the results obtained, the hybrid DNN approach demonstrate an overall accuracy of 97% (95% CI: 85% - 99%) in an independent cohort of 34 samples, not used in the development of the DNN.
The reported correlation is strong (r> 0.99), between CLL cells as a percentage of total white blood cells, and the DNN and expert analysis.
The proposed method reduces the data processing time from 15 min to 12 s.
I have some reviewer notes:
Abstract. You have to show one sentence about how your work will be continued.
Response: We have now concluded the abstract with the following: "The proposed DNN demonstrated high accuracy in the CLL MRD detection and significantly improved workflow efficiency. Additional clinical validation is needed before it can be fully integrated into the existing clinical laboratory practice." Page 1, line 31-33
Introduction. This part is too short. You have to give more details about what authors have as a results in their works.
Response: We have included additional relevant background on computational and automated flow cytometry on page 2, line 66-83.
“2.3. Deep neural network (DNN)”. It is not clear how you select the sample size.
Response: Currently, there is no standard or widely accepted approach for determining an appropriate sample size for training of a neural network, since the sample size depends on the complexity of the problem and the complexity of the model/ architecture used. In the clinical laboratory, samples for algorithm training are usually uses non-probabilistic purposive sampling of most or all available cases due to cost and other logistic consideration. For this study, purposive sampling of all available clinical cases between the years 02/2020 and 05/2021 were used in the development cohort. We have included this information on page 2, line 91-92.
“2.4. Statistical analysis”. What is your level of significance?
Response: Hypothesis testing and confidence intervals used an alpha of 5%. We have indicated this on page 4, line 178: “The level of significance (α) was chosen as 0.05.”
Table 1. Are the accuracy levels dimensionless? You have to describe it in the text above the table. Also for Table 2.
Response: We have indicated in Tables 1 and 2 that “Accuracy, Sensitivity, Specificity, PPV and NPV are presented as dimensionless ratios.”
Figure 2B. It is not clear why there is low correlation between dependent and independent variables in your linear model. Where is the problem?
Response: Performance degradation of the F-DNN compared with L-DNN at abnormal CLL counts <1000 could be an artifact of a significantly higher number of abnormal CLL events used in the F-DNN training cohort compared with the L-DNN training cohort. For the F-DNN, the training cohort included 10,624,335 abnormal CLL cells, while for the L-DNN, the training cohort included 13,182 abnormal CLL cells. This resulted in an apparent measurement bias of F-DNN towards “over-calling” abnormal CLL events (false-positive events), whereas the L-DNN predicted fewer false-positives compared with the expert gating. The hybrid DNN approach serves to mitigate against measurement bias due to class imbalance by combining both the F-DNN and L-DNN in a sequential fashion.
Figure 4. Are all data in the confidence interval?
Response: The 95% confidence intervals (CI) for the slope and intercept from Passing-Bablok regression were obtained from bootstrap resampling and the CIs are very narrow and do not deviate substantially from the dashed line in Figure 4 to be visually apparent. The 95% CI for the regression line indicates that there is a 95% probability that the true best-fit line for the population lies within that confidence interval and does not need to include all data points. The reviewer may be referring to a concept called the “prediction interval” here, where one looks at any specific value of x (e.g. x0) and finds an interval around the predicted value Å·0 for x0 such that there is a 95% probability that the real value of y (in the population) corresponding to x0 is within this interval. The intention of showing the regression line is to demonstrate the equivalence of the two methods and not to demonstrate what the prediction interval would be for a new value.
Discussion. You have to compare your results with minimum 3 other papers (More is better). Also, you have to show what is the accuracy of the results in other papers.
Response: Although automated flow cytometric MRD assessment has been reported in B-ALL and AML (Reiter M et al. Cytometry A 2019, 95, 966-975 & Weijler L et al. Cancers (Basel) 2022, 14, doi:10.3390/cancers14040898.), it is not possible to directly compare the result of our accuracy to other flow cytometry models due to distinct diseases, distinct flow panels, the complexity of the problem, the difference/complexity of the AI model architecture used. Instead, we mentioned the automated MRD assessment in the introduction.
Conclusion. You have to show how your results improve the known solutions in this study area.
Response: To our knowledge, this is the first study to prove the accuracy and efficiency of AI in the quantitative detection of CLL MRD, which broadens the application of AI in disease monitoring and prognostic prediction (page 16, line 396-398). However, we cannot conclude that our model improves the known models in this area because a side-by-side comparison is needed for that intent, but it is beyond the scale of our study.
I have some suggestions:
It will be good to show the accuracy of the compared methods, presented from other authors. Also, make more comparative analyses. It will improve your contribution.
Response: We appreciate this suggestion. However, as mentioned above, we cannot perform a direct comparison with other models because we are studying different diseases and using different models.
Reviewer 3 Report
Comment 1.
The authors established the deep neural network(DNN) with the three fully connected hidden layers in order to chronic lymphocytic leukemia minimal residual disease(CLL MRD). The authors should explain why they had chosen the number of DNN hidden layers as 3, that is, the authors should show the comparison in terms of DNN output results when the number of DNN hidden layers had been set up as 3, 4, 5 and 6.
Comment 2.
The authors should show the comparison in terms of DNN output results as to which activation function (e.g., ReLU, softmax and so on) is more better for DNN output result.
Author Response
We greatly appreciate the comments of the reviewers and believe that the suggested additions to our manuscript have served to further enhance its quality.
Reviewer #3
Comment 1.
The authors established the deep neural network(DNN) with the three fully connected hidden layers in order to chronic lymphocytic leukemia minimal residual disease(CLL MRD). The authors should explain why they had chosen the number of DNN hidden layers as 3, that is, the authors should show the comparison in terms of DNN output results when the number of DNN hidden layers had been set up as 3, 4, 5 and 6.
Response: The choice of 3 hidden layers was based on earlier development work using a different use case [Camp, J. et al. Deep neural network for cell type differentiation in myelodysplastic syndrome diagnosis performs similarly when trained on compensated or uncompensated data. In Proceedings of the Medical Imaging 2022: Digital and Computational Pathology, 2022; pp. 205-214] and observations of another group that used principal components analysis to study the optimal number of hidden layers in a DNN [Choldun I, et. al. Determining the number of hidden layers in neural network by using principal component analysis. In Proceedings of SAI Intelligent Systems Conference 2019 Sep 5 (pp. 490-500). Springer, Cham.]. We have included this information in page3, line 129-132.
Comment 2.
The authors should show the comparison in terms of DNN output results as to which activation function (e.g., ReLU, softmax and so on) is more better for DNN output result.
Response: Softmax activation function and categorical cross-entropy are standard techniques used for multi-class problems, where the classes are mutually exclusive [Grandini M, Bagli E, Visani G. Metrics for multi-class classification: an overview. arXiv preprint arXiv:2008.05756. 2020 Aug 13]. ReLU enables the more efficient training of deeper networks, compared to the commonly-used activation functions before 2011, for instance, the sigmoid or the hyperbolic tangent. ReLU is six times faster than other well-known activation functions and is one of the most commonly used activation functions in DNNs [Szandała T. Review and comparison of commonly used activation functions for deep neural networks. InBio-inspired neurocomputing 2021 (pp. 203-224). Springer, Singapore]. Studies have also shown that use of the Adam optimizer improves the performance of wide and deep neural networks [Jais IK, Ismail AR, Nisa SQ. Adam optimization algorithm for wide and deep neural network. Knowledge Engineering and Data Science. 2019 Jun 23;2(1):41-6.]. We have included this information on page 3, line 134-141.